# Macrophages and HLA-Class II Alleles in Multiple Sclerosis: Insights in Therapeutic Dynamics

**DOI:** 10.3390/ijms25137354

**Published:** 2024-07-04

**Authors:** Petros Prapas, Maria Anagnostouli

**Affiliations:** 1Research Immunogenetics Laboratory, First Department of Neurology, Aeginition University Hospital, School of Medicine, National and Kapodistrian University of Athens, Vas. Sofias 72-74, 11528 Athens, Greece; 2Multiple Sclerosis and Demyelinating Diseases Unit, Center of Expertise for Rare Demyelinating and Autoimmune Diseases of CNS, First Department of Neurology, School of Medicine, National and Kapodistrian University of Athens NKUA, Aeginition University Hospital, Vas. Sofias 72-74, 11528 Athens, Greece

**Keywords:** multiple sclerosis, macrophages, microglia, HLA, T cells, immune response, HLA-immunogenetics

## Abstract

Antigen presentation is a crucial mechanism that drives the T cell-mediated immune response and the development of Multiple Sclerosis (MS). Genetic alterations within the highly variable Major Histocompatibility Complex Class II (MHC II) have been proven to result in significant changes in the molecular basis of antigen presentation and the clinical course of patients with both Adult-Onset MS (AOMS) and Pediatric-Onset MS (POMS). Among the numerous polymorphisms of the Human Leucocyte Antigens (HLA), within MHC II complex, HLA-*DRB1*15:01* has been labeled, in Caucasian ethnic groups, as a high-risk allele for MS due to the ability of its structure to increase affinity to Myelin Basic Protein (MBP) epitopes. This characteristic, among others, in the context of the trimolecular complex or immunological synapsis, provides the foundation for autoimmunity triggered by environmental or endogenous factors. As with all professional antigen presenting cells, macrophages are characterized by the expression of MHC II and are often implicated in the formation of MS lesions. Increased presence of M1 macrophages in MS patients has been associated both with progression and onset of the disease, each involving separate but similar mechanisms. In this critical narrative review, we focus on macrophages, discussing how HLA genetic alterations can promote dysregulation of this population’s homeostasis in the periphery and the Central Nervous System (CNS). We also explore the potential interconnection in observed pathological macrophage mechanisms and the function of the diverse structure of HLA alleles in neurodegenerative CNS, seen in MS, by comparing available clinical with molecular data through the prism of HLA-immunogenetics. Finally, we discuss available and experimental pharmacological approaches for MS targeting the trimolecular complex that are based on cell phenotype modulation and HLA genotype involvement and try to reveal fertile ground for the potential development of novel drugs.

## 1. Introduction

Multiple sclerosis (MS) is a chronic, autoimmune, and neurodegenerative disorder characterized by increased acute neuroinflammation within the central nervous system (CNS), resulting in demyelination, axonal damage, and ongoing disease progression. The disease etiology relies on a multifactorial pathological background involving genetic as well as environmental factors and represents a significant challenge in modern medicine due to the vague mechanisms that take place around onset and progression [1]. In a nutshell, MS is defined by the presence and activity of auto-reactive T-cells against myelin in the periphery; as such, infections or allergies can cause the implementation of molecules that show molecular similarity with some CNS antigens, a process also known as molecular mimicry [2]. Down the line, those cells trigger the differentiation into T_h_1 or T_h_17 (T-helper cells) leading to the onset of inflammation and activation of large signaling contributors (IFN-γ, TNFα, IL-17, IL-22, IL-21) [3,4]. The exact events that can lead to the onset of MS remain unclear to this day as available literature mainly focuses on the impediment against the disease progression rather than its onset and prevention. The first thing to investigate should be the detailed interactions between first line antigen presenting cells (APCs) and T-cells providing the initial signal that causes malfunction downstream of the immune response. T-cell functional activation requirements consist of three individual signals: (I) interaction of the T-cell receptor (TCR) with the antigen bound in the surface of APCs through the major histocompatibility complex II (MHCII), (II) activation of surface CD28 of CD4^+^ and CD8^+^ cells by monocyte derived CD80^+^ and CD86^+^, and (III) the effect of interferon γ (IFN-γ) and interleukin 4 (IL-4), also responsible for manipulating macrophage and microglia plasticity [5]. As neuroinflammation is significantly characterized by disrupted macrophage and T-cell state-homeostasis, it is possible that modifications in the cascade of polarization could drastically change the course of neurodegeneration in MS, Neuromyelitis Optica Spectrum Disorder (NMOSD), Myasthenia Gravis, etc. [6].

In the context of MS, whose etiology and progression depend significantly on the disruption of innate immune homeostasis, it becomes imperative to direct attention toward APCs such as macrophages and monocyte-derived populations. Valuable investigations, including in vitro and ex vivo studies of BV-2, THP-1, and other macrophage-like lines, have directly or indirectly manifested manipulation of polarization, shifting the spotlight onto the unexplored disruptions in the delicate phenotype equilibrium regarding MS and neuroinflammation [7,8]. The “old school” discrimination of macrophages through the M1/M2 paradigm might be a strong example that immune cell polarization is not often binary but a continuous procedure. As we have observed in numerous other diseases (autoimmune, fibrotic, inflammatory, cancer, etc.), macrophages and their functional subsets play a pivotal role in orchestrating disease progression by modulating antigen presentation, cytotoxicity, auto-reactivity, and influencing wound healing [9,10,11]. A striking illustration of this paradigm can be found in autoimmune rheumatoid arthritis (RA) [12]. However, in MS, it is the pro-inflammatory profile that emerges as the primary instigator of the disease pathology both for macrophages and dendritic cells (DCs) [13]. Newly characterized APC populations emerge constantly regarding tissue specific populations and even systemic myeloid lines, aiming to shed light on the functional image of various diseases [14,15]. Subsets such as M2a, M2b, and M2c are characterized as independent functional groups, each serving unique functional roles that contribute to the complexity of the immune response [16,17].

One pivotal remark in autoimmunity research is the decomposition of MHC II function. Due to its mediating role between innate and adaptive immunity and the substantial polymorphisms found in its genetic locus, MHC II forms an intuitive link between APCs and CD4^+^-mediated response in two distinct ways. First, MHC II presence and expression has been proven important for the maintenance of immune cell populations in rest conditions. Moreover, TCR reshaping based on genetic or environmental factors often create a vital bedrock for autoimmunity to emerge; MHC II structure often favors the spontaneous presentation of autoantigens resulting in the rapid cascade signaling that leads to perpetual auto-reactivity [18]. MHC II also plays a role in the regulation of autoantigen tolerance by sparing or depleting autoreactive thymocytes [19]. In MS, as we are going to discuss, MHC II seems to play a significant role both as risk for creating the initial trigger as well as acting as the catalyst for passing and amplifying this signal down. The MHC locus houses a highly complex network of 224 functional genes that are densely packed and closely interconnected, both in terms of their anatomical proximity and biochemical activity [20]. It has been demonstrated that small alterations in the functionality of a single gene within this network has the potential to affect the integrity of other genes. One particular example is the tumor necrosis factor (TNF) gene family, located within the MHC locus, which plays a substantial role in regulating inflammatory processes [21]—polymorphisms within the TNF-α can result in modification in the expression patterns of separate functional genes within the locus [22]. This further underscores the multifaceted and interconnected nature of MHC II’s role in the context of MS and broader immunological disorders through epistatic mechanisms within the MHC region, among others [23]. Within the vast genetic landscape of the MHC gene locus, a pivotal gene family claims the spotlight: HLA class II genes. Researchers have directed their attention to these genes due to their potential to significantly influence disease risk through genetic mutations or the presence of specific alleles. The range of elements influenced by these genetic factors is extensive, encompassing infectious diseases like COVID-19, idiopathic conditions such as fibromyalgia and Alzheimer’s, and autoimmune disorders like MS. Notably, the *HLA-DRB1*15:01* allele has come under intense scrutiny in population studies given its notable presence in a range of diseases, including acute disseminating encephalomyelitis, ulcerative colitis, MS, Epstein-Barr virus infection (EBV), COVID-19, systemic lupus erythematosus (SLE), and non-infectious respiratory diseases, among others [24,25,26]. While the function and clinical stigma of HLA alleles has been investigated for several diseases, the specific molecular underpinnings in the context of MS remain a glaring gap in the existing body of knowledge. This gap underscores the compelling need for in-depth research into the molecular basis and biochemical functions of the *HLA-DRB1*15:01* allele. The functional characterization of some unpopular risk alleles such as *DRB1*04:01* and *DRB5*02:01* could provide a new perception about the development of neurodegenerative diseases, and MS in particular, setting the basis for novel treatments to arise.

This narrative review aims to comprehensively describe the effect of HLA class II alleles on MS and highlight the core and multifactorial function of macrophages and microglia as core drivers in numerous aspects of the disease onset and progression. We are also going to assess alternative activation pathways of APCs, suggesting some novel interactions that might play crucial role in disease onset and progression in MS, as well as in other autoimmune diseases. Additionally, we will explore the molecular pathways involved in immune cell migration during Blood-Brain Barrier (BBB) breach in MS, highlighting the points of interest in antigen presentation. Finally, we will discuss how current pharmacological approaches utilize MHC II and physiological functions of macrophages and microglia, underscoring some special interactions, in particular HLA-DRB alleles, and systemize applied and innovative strategies to refine therapeutic interventions in MS. We believe that a thorough understanding of these immune mechanisms and their genetic background holds promise of advancing valuable precision medicine approaches for managing neurodegeneration in autoimmunity.

## 2. HLA Class II Risk Alleles

The MHCII region contains the HLA class II molecule, a transmembrane protein consisting of two chains, DRA and DRB. The general HLA complex is located in the short arm of chromosome 6, and it is encoded by 6 exons, each responsible for a functionally different part of the protein [27]. Within the DRB chain, hundreds of polymorphisms linked to the peptide binding site have been discovered and many of them have been associated with increased susceptibility to certain diseases.

The *HLA-DRB1*15:01* allele has been identified as a major genetic risk factor for MS as well as in other diseases such as Myasthenia Gravis in European populations [28]; individuals who carry this allele have been shown to have a significantly increased risk of developing pediatric-onset MS (POMS) and adult-onset MS (AOMS) [29]. Furthermore, regarding AOMS, *DRB1*15:01* carriers have been shown to exhibit onset of the disease earlier than non-carriers [30]. On a relative axis, *DRB1*15:01* has been associated with familiar-MS, appearing more frequently in 2nd and 3rd subgroups than in the 1st one [31].

The molecular mechanisms regarding *HLA-DRB1*15:01* risk of developing MS include both structural and expressional alterations. *HLA-DRB1*15:01* risk properties are correlated with the presence of a unique alanine residue at position 71 of the DRβ chain (DRβ71), which creates a large, hydrophobic P4 pocket in the peptide binding groove of the molecule. This structural similarity is also present in other alleles (*DRB1*15:06* and *DRB1*13:09*), which are found at a less significant rate in the general population [32]. Variations in the peptide-binding groove of *DRB1*15:01* may result in inappropriate immune responses and spontaneous presentation of MBP, for instance, by MHCII^+^ cells [33]. Furthermore, citrullination of MBP makes it vulnerable to post translational modifications by cathepsin D and matrix metalloproteinase-3 (MMP-3), resulting in alteration of the microstructure and affinity to the MHCII socket (or certain HLA-DRB components) and presumably playing a role in early neurodegeneration in MS [34]. Other identified post translational modifications of MBP can also result increase MHC affinity, for instance acetylation (Ac 1–11) has been shown to promote pathogenic T-cell excitation in mouse models [35]. A similar molecular background has been described for *DRB1*04:01*, which has been found to bind and display MBP_111–129_ with a high affinity [36,37]. The *DRB1*15:01* allele has also been experimentally associated with higher MHCII expression, which is perceived as a leading cause of grey matter lesions [38], suggesting a mediating neuroinflammatory function rather than direct impact on the molecular disease pathophysiology. Alleles that adopt a protective role against MS have also been suggested to function likewise by adopting altered molecular patterns. For example, *DRB1*01:01* owes its protective role to its affinity with the non-classical peptides MBP_152–161_ and MBP_90–98_ resulting in lower binding rates and the possible kinetic discrimination between exogenous peptides and endogenous MBP [39]. Notably, some risk-labeled alleles seem to carry a similar peptide sequence that has been linked with increased susceptibility to autoimmune diseases including MS. This shared epitope (SE) can be observed in *DRB1*15:01*, *DRB1*01:01*, and some *DRB1*04* alleles that we will discuss later on (see Section 5.2) [40].

*DRB1*15:01* is often found in combination with *DQB1*06:02* and *DQA1*01:02* alleles [41]; this combination are risk markers for MS even though their functional role has been controversial. However, the primary MHC II effect in the paradigm of MS does not seem to be mediated by variants in the *DQB1* and *DQA1* loci, but the *DRB1*15:01* locus instead [42]. Additional MS risk alleles for MS include *DRB1*13:01*, *DRB1*08:01*, *DQB1*03:02* [43], *DPB1*03:01* [44], and *DPB1*04:01* [45]. Regarding those alleles, independent MHC effects have been described for *DRB1*15:01*, **03:01*, and **13:03* [29]. *DRB1*03:01*, an allele strongly linked with *HLA-DQB1*02:01* [29,44], has been proven to be a susceptibility factor for POMS and neuromyelitis optica among Caucasian cohorts [43,46].

In POMS, the involvement of innate immunity and HLA alleles in the development of the disease is highly speculated but not fully understood. Apart from *DRB1*15:01*, POMS has also been associated with increased presence of *DRB1*03:01* in some cohorts, as well as with decreased annualized relapse rates when compared with AOMS [47,48]. Older studies on Greek cohorts demonstrated that patients with POMS had significantly more acute relapses and more thoracic spinal cord lesions when they carried the *DRB1*03* allele [49]. Neuroimaging findings implemented in a recent study by our facility, however, describe an association of *DRB1*03* with reduced risk for brainstem lesion development in POMS patients, further complicating the landscape of innate allele-mediated risk [50]. Additionally, risk to the disease has been associated with *DRB1*08*; however, its effect is significantly lower than *DRB1*15* and *DRB1*03* [51]. Finally, complementing the significant correlation between the *DRB1*15* allele in both adult and pediatric-onset MS, newer findings hint a potential role of *DRB1*16* as a protective factor in POMS and *DRB1*11* for AOMS, warranting further investigation to fully elucidate these genetic predispositions [52].

## 3. Macrophage Ontogeny in MS

In the neurodegenerative CNS, there are four main macrophage populations observed: resident microglia, perivascular macrophages (pvΜΦ), leptomeningeal macrophages (lmMΦ), and choroid plexus macrophages (cpΜΦ) [53]. Microglia is the main subset that resides in the CNS in rest conditions and surveils the brain parenchyma for pathogens. In their research around macrophage ontogeny and cell cycle, Guilliams et al. describe resident macrophages as “guards”, each surveilling a defined area within the tissue they reside in, mentioned as a ‘niche’ [54]. Cells deriving either from circulation or yolk sac lineage occupy empty niches, creating a consistent formation. Under rest conditions, CNS niches are occupied by resident “resting state” microglia derived from myeloid progenitors such as yolk sack monocytes. Microglia, unlike other tissue-specific resident macrophages, receive minimum to no replenishment from the circulation from CX3CR1^+^Ly6C^+^ monocytes and maintain their population and activation status through self-renewal [55,56] independently of the serum M-CSF levels [57]. They are characterized by the presence of conventional markers CD11b, CD14, and IBA-1; however, they are specifically distinguished from peripheral macrophages by surface purinergic receptor P2RY12 and transmembrane protein 119 (TREM119) [13,58]. In MS lesions, TREM119 is a valuable tool in discrimination between microglia and infiltrating macrophages, even though its levels are shown to decrease after activation [59]. Interestingly, microglia associated with active demyelination have been shown to acquire a “foamy”, anti-inflammatory phenotype due to increased cholesterol and lipid uptake [60]. Moreover, microglia provide a significant HLA-DR signature in the CNS with HLA-DR production, correlating with the level of active neuroinflammation and consequently being elevated in acute lesions and active demyelination sites; in close association with this, microglia have been shown to elevate HLA-DR expression in reaction to paracrine IFN-γ stimulation and other neuroinflammatory conditions in MS [61]. In chronic inactive lesions, characterized by isomorphic demyelinated gliosis, there was a low density of microglia. These cells exhibited an amoeboid phenotype, expressing relatively low levels of HLA-DR but high levels of IBA-1 [62].

LmMΦs, a separate population of microglia-like cells, are the less characterized subset and are often grouped with other subsets and mentioned as subdural macrophages. lmMΦs strongly express CD68 and CD163 as well as IBA-1, but, unlike microglia, they have been shown to specifically express epidermal growth factor receptor (EGFR). Among this subset, a population of cells that grows independently of CSF1 has been identified whose functional role remains undescribed [63]. Most accepted studies indicate that pvMΦs are found in the perivascular space of the vessels across the CNS. In detail, pvMΦ presence has been reported between the smooth muscle layer of the pial arterioles and the borders of the brain parenchyma [64]. Alike microglia, they check positive for CX3CR1 and IBA-1 but they can be distinguished by the lack of purinergic receptor P2RY12 [65]. The cpMΦs, the macrophages found in the area around the choroid plexus and the epiplexus, are the main functional infiltrating population during neuroinflammation. They are the only subset that is actively replenished by circulating monocytes, suggesting high CX_3_CR and Lyve1 presence. The cpMΦs also show shifts in expressions relative to cell and host age; for instance, CD74 shows an increase in adults while CD206 expression is decreased [66].

Relatively, microglia expresses a low MHCII signature, therefore HLA-DRB1 allele differentiation might not play a crucial role in their direct function [65]. A higher HLA-DR signature has been observed selectively in the subventricular zone and thalamus [67]. In contrast, pvMΦ and cpΜΦ are characterized by high MHCII surface levels, suggesting a strong connection between HLA function and restimulation of memory CD4^+^ and effector T cells [68,69]. In the general cpMΦ population, MHCII may follow the opposite age-dependent pattern than CD206, with activation state homeostasis shifting towards the pro-inflammatory phenotype, as displayed in mice [70]. However, studies on EAE mice have shown that MHCII^+^ macrophages and microglia are not essential for the development of the model’s pathogenesis. These data may turn the spotlight on other APC populations in the inflammatory CNS, like dendritic cells for the instance of reactivation of myelin-reactive T-cells (Figure 1) [71,72].

In smoldering lesions, a type of mixed active-inactive lesions, activated macrophages and microglia are found on the edges (rim) of the inflammatory sites, contributing to slow lesion expansion and destruction of the surrounding parenchyma [73,74]. Smoldering lesion macrophages tend to acquire a consistently activated inflammatory phenotype, lacking a resting state while maintaining a high MHCII and CD38 signature, suggesting a strong DRB1 involvement. Furthermore, rim macrophages often undergo significant metabolic alterations involving altered mitochondrial functions and shift in utilization of nutrients such as glucose and glutamine [75]. A high iron signature is also observed in those cells, which can be associated with phagocytosis of iron-rich oligodendrocytes by aged microglia and senescence of the innate immune system [76].

## 4. APC Recruitment and T-Cell Activation through the Blood-Brain Barrier

During neuroinflammation, different types of APCs in the CNS activate T-cells through distinct pathways. Microglia, the resident macrophages in the CNS, present antigens to T-cells through the classical MHC class I and II pathways [77]. Astrocytes, which provide structural and metabolic support to neurons, can also present antigens to T-cells, although the pathways involved are less well understood [78]. Evidence suggests that oligodendrocyte apoptosis and microglia activation are the early steps for developing MS plaques and signal T-cell infiltration [34]. Astrocyte-mediated T cell activation can only occur following priming by microglia while some claim that B7-independent antigen presentation by astrocytes can induce apoptosis in CD4+ T-cells [79,80].

In the classical MHC class II pathway, microglia and other APCs present peptides derived from endocytosed extracellular proteins to CD4+ T-cells. The presentation of antigens by MHC class II molecules on the surface of APCs is enhanced by co-stimulatory molecules, such as CD80 and CD86 in microglia but not in astrocytes [81,82]. The activation of CD4+ T-cells can lead to the differentiation of effector T-cells that secrete pro-inflammatory cytokines and sustain the neuroinflammation [83]. In contrast, the MHC class I pathway presents peptides derived from endogenously synthesized proteins to CD8+ T-cells, which can recognize and kill infected or abnormal cells [84]. In the paradigm of MS, the main target of APCs is suspended myelin particles, which vary in shape and volume depending on the fragmentation patterns or the morphology of the target axon bodies. In this case, MHC II pathways are favored in comparison to MHC I, in which APCs struggle to succeed, sometimes leading to triggering of early apoptosis.

As CNS is not an enclosed environment, macrophage activation can be triggered through the BBB. The BBB is a semi-permeable barrier that separates the vascular system and the CNS and it is characterized by capillary endothelial cells forming tight junctions as well as a number of cellular components (receptors, transporters, efflux pumps, etc.) that control the influx of molecules [85,86]. Astrocytes that reside near the capillaries extend their end-feet to provide structural support to the vessels and regulate the permeability and homeostasis of the barrier via the release of cytokines [87]. Passive crossing of the BBB is generally possible only by lipophilic, positively charged molecules with a molecular weight from 400–600 Da [88]. Active crossing of essential nutrients and other non-lipophilic molecules can be achieved through specific carrier proteins such as the glucose transporters (GLUT1), amino acid and monocarboxylate transporters, as well as receptors that perform transcytosis of proteins like insulin and transferrin [87]. Novel invasive techniques aiming at drug delivery involve BBB penetration through focused ultrasound with microbubbles, osmotic opening, and engineering of nanoparticles and liposomes to exploit innate transport mechanisms [89,90,91].

The subtypes of T_h_1 and T_h_17 cells are considered able to invade the CNS parenchyma through both the choroid plexus and the capillary breach in response to LFA-1/ICAM-1 receptor [92,93]. In the CNS, T_h_1 cells secrete high volumes of TNF-α and IFN-γ, attracting naïve monocytes and macrophages towards the breach sites, while T_h_17 cells co-expressing RORγt and T-bet can convert to T_h_1-like cells (T_h_17.1) in response to the high levels of IL-12 and IL-27 that characterize MS [94]. A double function of IL-27 as both pro-inflammatory and anti-inflammatory has been exhibited in EAE mice where continuous infusion of IL-27 resulted in improvement in disease progression while knock-out of the respective receptor mediated disease severity [94]. On the other hand, when produced in lower quantities in the CNS by microglia and astrocytes, IL-27 is perceived to promote a pro-inflammatory phenotype via STAT mediated TLR4 production in naïve monocytes but not in activated macrophages [95,96]. However, it has been observed that IL-27 has significant co-stimulatory function in LPS polarized macrophages, further supporting the positive feedback pro-inflammatory activation model [97]. The T_h_17 cells, generally known to possess anti-inflammatory and regenerative action through their producing cytokines, contribute to the demyelinating inflammatory phenotype through two distinct pathways: (1) Production of IL-17A and IL-17F in large quantities induces the downstream expression of pro-inflammatory cytokines and chemokines in macrophages, epithelial, and endothelial cells and modifies migration patterns of neutrophils [98]. Furthermore, IL-17A has been observed to break down BBB in RRMS patients; and (2) IL-17 expression in astroglia is increased in EAE and excessive IL-17R ligation results in secretion of pro-inflammatory cytokines and destabilization of the BBB through astrogliosis [99,100]. IL-21 produced by T_h_1 cells exhibits paracrine auto-activation properties in neighboring T_h_17 and naïve CD4^+^ cells, generating a positive feedback loop amplifying previously mentioned functions [96]. During neuroinflammation, the recruitment of macrophages to the CNS is a critical component of the immune response. Macrophages can be recruited from the circulation across the BBB, which is formed by specialized endothelial cells and is impermeable to most immune cells [77]. The recruitment of macrophages is facilitated by the expression of adhesion molecules on the endothelial cells and the production of chemokines by resident cells in the CNS. Once in the CNS, macrophages turn to a highly reactive state, producing large volumes of pro-inflammatory cytokines and therefore sustaining the inflammation and contributing to the retention of breach sites by astrocyte destabilization.

Under normal conditions, the CSF is dominated by monocyte-attracting chemokines, mainly CCL-2 (MCP-1), IL-8, fractalkine, MIP-1β, MIP-1δ, and ΜΙP-3^α^ [101]. When the BBB is weakened and breach sites begin to appear, CCL2, CCL3 (MIP-1α), and CCL5 (RANTES) are able to take effect on myeloid cells such as naïve macrophages and DCs and, in some cases, specific B-cell populations, resulting in modulation of their normal migration patterns [102,103]. By binding to their respective receptors on blood monocytes and macrophages, these chemokines facilitate the migration and recruitment to sites of BBB disruption. The cells then crawl along the endothelial surface until they reach a site where the BBB is disrupted, such as an area of inflammation or injury, and then they squeeze through the endothelial cells and enter the CNS parenchyma. Once in the CNS, macrophages can interact with T-cells as well as with resident cells, such as microglia and astrocytes [104].

In patients with MS, a striking and persistent pattern of inflammation and acute immune activation within the choroid plexus (CP) has emerged from radiological and immunostaining investigations. Remarkably, this immune activity appears to persist over extended durations, maintaining a progressive profile, even during phases when neurodegeneration is believed to predominate [105]. The CP resembles an inflammatory landscape similar to that of acute viral encephalitis, with a prominent presence of HLA-DR activity in both the CP and stromal regions. Storminger et al., in a relevant study, describe the CP as an active niche for antigen presentation cells, facilitating a large variety of MHCII^+^ cell populations, such as CD11b^+^, CD11c^+^, CD68^+^, and CD20^+^ as well as an undescribed myeloid subset that is characterized as CD45^high^CD11b^+^CX3CR1^−^MHCII^+^ CD11c^high^ [106]. This heightened HLA-DR activity is a consequence of the influx of peripheral blood macrophages, which successfully traverse the BBB to access the CSF space, homing in on demyelination hotspots. For example, in Kholmer or epiplexus cells, an important subset of MHC-II^high^ cpMΦs, macrophages breach the Blood-CSF barrier under inflammatory conditions by utilizing CSF draining through the leptomeningeal space towards the CNS [107,108]. Kholmer cells, among other MHC-II^high^ populations, are theorized to be significantly affected by the *HLA-DRB1* genotype, resulting in alterations of inflammatory patterns; however, new studies should be conducted to provide further support to this statement. While dendritic cells (DCs) are theorized to exhibit robust antigen-presenting capabilities, specific DC markers are not abundantly found in the CP in human studies.

Complementing this landscape, antigen-presenting endothelial cells of the BBB and choroid plexus epithelium demonstrate a robust expression of the integrin ligands intracellular adhesion molecule 1 (ICAM-1), and vascular adhesion molecule 1 (VCAM-1), which regulate cell migration by modulating adhesion, akin to their role in inflammatory macrophages [109]. Circulatory T-cells also squeeze through the breach sites into the CSF in a process exclusively mediated by ICAM-1 and ICAM-2 reaction, with lymphocyte function associated antigen 1 (LFA-1) found in the surface of T-cells [110]. Sites with a low density of adhesion molecules or impaired endothelial functions obstruct the transition of some cells through the BBB breach. Furthermore, another study described some interesting transition patterns of CSM B-cells through selective CP permeability in MS patients, resulting in the presence of the potentially harmful cells in the CSF of the patients [111]. On top of that, a steady and high influx of naïve T-cells, the main lymphocyte population found in the CP, is sustained in low inflammatory states, finding their way into the CSF and taking part in the circular activation scheme, rapidly replenishing autoreactive T-cells [112]. Notably, the inflammatory conditions persist within the CP, even during periods of relative inactivity in patients with RRMS [113]. This consistent presence of inflammation paves the way for a positive feedback loop, where inflammatory macrophages maintain an open breach, facilitating a continuous influx of these immune cells from the periphery into the CSF. This process ultimately sustains a high feed of auto-reactive T-cells within the CSF, further stimulating naïve and polarized macrophage reactivity resulting in a perpetual activation cycle and therefore highlighting the complex dynamics of MS pathogenesis regarding macrophages and innate immunity in human studies (Figure 2).

## 5. Macrophages to T-Cell Activation and HLA Class II

### 5.1. Macrophage Activation

In general, M1 macrophages are known to produce pro-inflammatory cytokines such as IL-1β and TNF-α, which stimulate the activation and proliferation of CD4+ T cells. Then, CD4+ T cells mediate the immune response in MS and are thought to be involved in the destruction of myelin [114,115,116]. On the other hand, M2 macrophages have been shown to inhibit T_h_1/T_h_2 signaling and promote tissue healing and resolution of neuroinflammation. M2 macrophages boost production of anti-inflammatory cytokines such as IL-10 and transforming growth factor beta (TGF-β), which can suppress the activation and proliferation of CD4+ T cells [117]. M2 macrophages also play a significant role in tissue repair and may promote the regeneration of damaged myelin [114,115,118].

The increase of the M1 phenotype prior to and during the pathology of MS and the enrichment of the CSF with inflammatory cytokines, such as IL-1β and tumor TNF-α, lead to the rapid activation and proliferation of CD4+ T cells. M2 microglia/macrophages, however, contribute to the reconstruction and remodeling of the myelin scaffold, with many novel therapies targeting the shift of polarization towards this state. A study in cerebellar slice cultures demonstrated the ability of M2-derived activin-A to control oligodendrocyte differentiation during remyelination [119]. On the same axis, experiments in rat primary glial cells exhibited the autocrine and paracrine ability of some TGF-β isoforms to induce oligodendrocyte differentiation [120]. Despite a plethora of studies on macrophage polarization providing a well-detailed picture around inflammatory M1 and tissue remodeling M2 macrophages, recent findings suggest a much more complex scene. IFN-γ, LPS, and TNF-a activated M1 macrophages exhibit a more straightforward functional domain and are involved in type I inflammation, killing of intracellular pathogens, and tumor resistance.

Unlike the straightforward function of the M1 subset, M2 macrophages are suggested to acquire a multi-dimensional activation pattern involving three subcategories: M2a, M2b, and M2c (or M2o). The M2a subset is stimulated by IL-4 and IL-13 via IL4R activation, which leads to alternative transcriptional patterns through JAK1/JAK3 and STAT6 activation. It is characterized by surface IL-10, IL-1a, CD206, and Polyamine and its main functions include tissue repair, signaling of T_h_2 associated responses such as type II inflammation and MHCII regulated phagocytosis, and antigen presentation [121]. The M2b subset is activated by TLR and IL-1R ligands and has IL10^high^IL12^low^ and MHCII^high^ CD86^+^ phenotypes. M2b are considered to derive from IgG-Fcγ dependent B-cell responses activated by TLR signaling and they produce TNF and IL-1. M2b microglia resemble the M1 most due to their increased expression of IL-6; they also have IL10^high^IL12^low^ and MHCII^high^ CD86^+^ phenotypes, which suggest their potential to stimulate T-cells [122,123]. Production of M1-associated cytokines is attenuated by JAK1 activation, which leads to translocation of STAT3 [124]. Despite its resemblance to the M1 phenotype and CD86 and MHC/HLA-DR presence, their general function and population dynamics are not always aligned. For instance in AD, where, like MS, CNS macrophages are skewed towards inflammatory populations, it has been described that M2b macrophages tend to decrease upon M1 increment [125]. Finally, the M2c subset, often referred to as M2o due to its “inactivated”, non-inflammatory profile, is believed to be activated solely by IL-10 and glucocorticoids. Its functional role suggests involvement in matrix deposition, versican-regulated cell adhesion, and migration and tissue remodeling in scaring and trauma [16,17,126]. M2c cells are characterized by increased expression of TGF-β, CD163, and sphingosine kinase (SPHK1), as well as the surface marker SLAM. Among those, TGF-β and SPHK1 expression is strongly associated with MS, suggesting a potentially significant role in the pathophysiology of the disease [127].

### 5.2. The Importance of MHCII 

Today, increased evidence emerges supporting that MHC II may play a key role in the behavior of macrophages as the presence or absence of specific alleles alters the extent of naïve cell activation (look at *DRB1*04:01* paradigm) [128]. Therefore, better characterization and classification of the cells based on their MHCII associated behavior is a necessary step in the investigation of HLA-DR and macrophage association. Despite the indications of HLA-DR’s involvement in macrophage polarization, studies targeting the topic directly remain limited. In Tumor Associated Macrophages (TAMs), a correlation has been shown between HLA-DR presence and the induction of M2 phenotype; however, the effect of different alleles has not yet been investigated [129]. These results, when combined with the proven allele-dependent modulation in expression levels, suggest the existence of a similar mechanism in MS that fits our investigation criteria and is a perfect candidate for future research. To further support this hypothesis, a recent study on transgenic mice highlighted the potential of HLA alleles in regulation of neuroinflammatory events through macrophage activation. Van Drongelen et al. showed that the *DRB1*04:01* risk allele promoted the M1 activation pattern in conventionally activated bone marrow derived macrophages (BMDMs) while *DRB1*04:02* polarized cells moved towards the M2 phenotype [128]. The importance of this study lies in the fact that the two alleles differ in only three amino acid residues, suggesting the high functional importance of HLA downstream of the antigen-presentation pathways. The *DRB1*15:01* and some other MS risk alleles were not positively associated with polarization in this study as they do not carry the researched RA-risk SE allelic epitope. *DRB1*04:01* has been shown to alter Tregs function, which is associated with maintenance of immune tolerance and prevention of autoimmunity [36]. Moreover, it has been shown to play a protective role in Parkinsons disease, Alzheimer’s disease, RA, and has also been suggested as a protective allele against Systemic Lupus Erythematosus (SLE) [15,130].

Recently, a connection has been established between the function of the HLA complex and the expression of the negative immune checkpoint regulator V-Domain Ig Suppressor of T-cell Activation (VISTA). VISTA is a regulatory molecule intricately associated with the presence of M2 macrophages; within the CNS, it is primarily expressed by monocytes and, to a lesser extent, by epithelial cells. Notably, VISTA expressed by monocytes keeps its active soluble form through proteolytic cleavage facilitated by MMP9, an M2-related factor [131,132]. Despite some evidence suggesting that VISTA expression decreases in microglia during lesions in individuals with MS, the existence of a positive correlation between risk alleles, such as *DRB1*15:01*, or polarization modulating alleles, such as *04:02*, has yet to be confirmed. Nevertheless, a speculative hypothesis can be proposed, wherein VISTA expression may serve as a mediating factor influencing M2 macrophage production, contingent upon the specific DRB1 allele carried by the individual.

### 5.3. TCR Function

Interactions between macrophages and T cells through antigen-presentation are perceived to contribute to the development of MS lesions, which are characterized by the infiltration of immune cells into the CNS and the destruction of myelin. By promoting T cell activation and TCR reshaping [92,133], M1 macrophages are actively found in all lesions and seem contribute to their formation and progression, while M2 macrophages are found mainly in chronic lesions and not in acute, playing a role in remyelination, oligodendrocyte activation, and prevention of further damage to the myelin [119]. A recent study suggested a strong association between TCR component CDR3 and five MHCII risk alleles, including *HLA-DRB1*15:01* [134]. CDR3, the most variable region in the TCR complex, interacts directly with the MHCII, leading to reshaping of the TCR receptor, affecting the responsiveness to presented antigens and potentially MS related autoantigens [133,135].

Dysregulation of CTLA-4, whether due to genetic factors or pathological mechanisms, is a common occurrence in MS. Studies have established a link between HLA alleles and autoimmune disorders, underlining, for instance, *DRB1*03* as risk factor for autoimmune diabetes regarding CTLA-4 downregulation [136]. Other than that, the risk alleles implicated in MS appear to exert a protective influence on the downstream regulation of CTLA-4 within the T-cell activation pathway [137]. When an APC presents to a T-cell, it provides signals through the MHC-TCR complex (signal 1) and co-stimulatory molecules like CD80/CD86 binding to CD28 on T cells (signal 2) [138]. This activation pathway can be modulated by CD40-CD40L interactions, which increase the expression of B7 molecules on APCs, therefore providing stronger costimulatory signals [139]. Supporting this paradigm, a promising phase II study with frexalimab is ongoing, targeting CTLA-4 downregulation in T cells through inhibition of CD40L with significant results [140]. The latter clearly shows the core role of the trimolecular complex in immune response and, in parallel, its core role as a central therapeutic target, with great further potential.

## 6. Epigenetic Modulation of *HLA-DRB1*15:01* in MS

Several studies have examined the effect of methylation in the expression of DRB1 genes and its potential effect in multiple disease pathophysiology. In general, it has been shown that within the MHCII locus, HLA-DRB1 expression is highly dependent on methylation patterns, with ~83% of the genes in the exon of DRB1 and ~85% in the intron showing significant correlation [141]. DNA methylation patterns in MS patients were significantly different from those in healthy controls and were associated with altered expression of several genes involved in immune regulation, including *DRB1*15:01* [142]. Furthermore, all altered differentially methylated regions (DMRs) were in the HLA class II region, presumably affecting genes like HLA-DRB5, -DRB1, -DQB1, -DQA1, -DQB2, and other MS susceptibility loci [143]. New studies keep revealing new patterns of the MHC methylome, suggesting causality between numerous single nucleotide polymorphisms and MS risk [144]. Another study found that histone modifications in MS patients were also significantly different from those in healthy controls and were associated with altered expression of *DRB1*15:01* [145]. Specifically, acetylation of histone H3 within the *DRB1*15:01* promoter region was associated with increased expression of the gene in MS patients, suggesting that epigenetic modifications of histones may also play a role in regulating *DRB1*15:01* expression in MS. In addition to DNA methylation and histone modifications, non-coding RNA expression has also been implicated in the regulation of *DRB1*15:01* in MS. One study found that expression of a specific microRNA, miR-326, was increased in MS patients and was associated with decreased expression of *DRB1*15:01* and other immune-related genes [146].

Altered methylation outside the HLA locus has also been identified in genes regarding cell cycle and metabolic processes. Apart from APCs, CD4^+^ and CD8^+^ cells have been shown to express differentially methylated phenotypes with alterations with four loci (MOG/ZFP57, HLA-DRB1, NINJ2/LOC100049716, and SLFN12) bearing interest for their involvement in core disease associated functions [147]. Hypomethylation specified for genes such as IL-13, IL-17, IFN-γ, and FOXP3, as well as hypermethylation of SHP-1, can also result in alterations of T-cell differentiation and/or increased inflammatory patterns driven by leucocytes [148]. Additionally, it has been suggested that upregulation of the enzyme Peptidylargininedeminase-2 (PAD2) can lead to destabilization of an MBP via citrullination and has also been shown to target vimentin on macrophages and actine in neutrophils, disrupting their homeostatic function [149].

On another axis, cross-referencing findings suggest a connection between sunlight exposure- related vitamin D and HLA-DR regulation [150]. Higher MS susceptibility of *DRB1*15:01* carriers in regions with limited sunlight has been a hot topic of discussion, with earlier studies suggesting the existence of a vitamin D-regulated mechanism that controls gene expression of the HLA locus. Supporting this theory, electrophoretic mobility shift assays showed that the vitamin D receptor specifically binds to the VDRE in the *DRB1*15:01* promoter, positively affecting transcription patterns [151,152]. Another novel epigenetic factor that has been proven to casually reshape macrophage polarization is the long noncoding RNA labeled GAS5. This molecule, tested in EAE mice, shifts the balance towards M1 by suppressing the transcription of topoisomerase-related function gene 4 (TRF4), a factor known to control M2 homeostasis [153]. Taken together, these studies suggest that epigenetic modifications may alter both the transcriptional patterns and the function of *DRB1*15:01* in MS patients, potentially contributing to the increased susceptibility to MS associated with this allele. Further research is required to uncover the mechanisms regarding these epigenetic modifications and their impact on *DRB1*15:01* function in MS.

It is important to note that interactions between epigenetic modifications and the assertion of their impact on disease progression remains a multifactorial and challenging task and focused research is required to discuss it further.

## 7. Pharmacological Approaches

The current therapeutical inventory against MS includes four first-line disease modifying treatments (DMTs): glatiramer acetate (Copaxone), intramuscular IF-β1a, subcutaneous IF-β1a and IF-β1b, teriflunomide, and dimethyl fumarate (DFM) [154,155]. Classical second-line DMTs in Europe are fingolimod and natalizumab; years ago, it also used to be mitoxantrone hydrochloride [156]. Many of those drugs have been shown to promote altered MHC II function and/or a shift in macrophage polarization directly or indirectly. The range of treatment options in the last seven years has opened up tremendously, and many new treatment options with new drugs are now available for patients with MS, each with different potential effects on the HLA system and vice versa. The effect of the HLA haplotype in the drug effectiveness and the interpretation of pathological interactions between CNS APCs and T-cells has been a critical research topic; however, no viable pharmacological approaches directly target the *DRB1*15:01* allele’s function yet [157]. The development of a targeted therapy towards high-risk alleles might offer valuable new means against both POMS and AOMS as well as late-onset MS (LOMS). Here, we discuss how some widely available drugs, or some in early development, function around our topic axis: MHCII mediated antigen presentation, macrophage polarization, and possible involvement of the HLA haplotype.

### 7.1. Proinflammatory Cytokine Inhibition

One applied approach is the targeting and regulation of inflammatory cytokines that dysregulate the innate immune response. For example, IFN-β1a-b and other drugs that inhibit TNF-α or IL-1β have been shown to reduce inflammatory stress and disease activity in MS [155]. HLA genes seem to play a role in modifying drug effectiveness in the case of IFN-β1a; one European cohort study revealed increased responsiveness by *DRB1*04* carriers, while in patients positive for *DRB1*15*, improvement of clinical outcome was less likely [158]. As IFN-β1a works by driving the core pro-inflammatory cytokine equilibrium down, the structural domain of those alleles might not be the direct reason for their functional distinction [155]. Additionally, a study on a mouse model suggests that silencing the transcription of c-Rel, a protein of the nuclear factor κΒ (Nf-κB) family that regulates lymphoid cell growth and survival, decreased secreted proinflammatory markers in macrophages and the subsequent induction of T_h_1 and/or T_h_17 responses [159,160]. Drugs, such as IFN-β1a and glatiramer acetate, that have strong immunomodulatory effects, can significantly reduce the activity of M1 macrophages and CD4+ T cells [155].

### 7.2. T-Cell Activation Targeting

Another approach is to target the activation of CD4+ T cells directly by macrophages and myelin antigens. One study suggests using antigen-specific therapies such as myelin peptides or monoclonal antibodies to induce immune tolerance to myelin antigens. Moreover, it has been speculated that down-regulating Nogo-A/NgR signaling could ameliorate the symptoms in MS patients by stabilizing the polarization balance of microglia [161,162]. On the same spectrum, infusion with anti-CD52 antibodies has been shown to downregulate the expression of MHCII and costimulatory molecules in infiltrating microglia and macrophages in the experimental autoimmune encephalomyelitis (EAE) model [163]. However, anti-CD52 therapies such as alemtuzumab do not exhibit a decrease in the total number of myeloid APCs in the mice blood [163]. Such therapies, in general, can shift the immune response towards a more anti-inflammatory profile and reduce the activation of M1 macrophages by driving CD4+ T cells activity down.

### 7.3. M2 Promotion

Therapies that promote the activation of M2 macrophages are proven to have a resolving effect in patients with MS. For instance, drugs that stimulate the production of IL-10 and TGF-β (such as IFNβ1 treatment) have been revealed to promote the activation of M2 macrophages and reduce inflammatory stress as well as tissue and nerve damage in RRMS and CPMS (chronic progressive MS) patients [3,164,165]. However, the anti-inflammatory shift might not be IFNβ1′s direct effect, as levels of M2 specific marker MMP9 do not seem follow a relative increase in the serum of patients who received the drug [166]. Next, glatiramer acetate, a drug that—among other functions—works by attenuating TNFα and IL-6 while enhancing TGF-β and IL10 has been shown to have significant effect in achieving NEDA-3 (No evidence of disease activity based on the absence of clinical relapse, disability progression, or radiological activity achievement) by verified shift in macrophage and myeloid suppressor cell plasticity [167,168]. FTY720 (fingolimod), a fairly popular immunomodulatory drug, directly suppresses M1 activation by spontaneous binding to the sphingosine 1-phosphate receptor (S1P) [169]. Although the molecular basis of fingolimod is not yet fully understood, one study on adipose tissue grafting suggests that it induces M2 polarization by intercepting STAT3 signaling [170]. Moreover, the lipophilic nature of the drug allows the drug to cross the BBB, therefore being able to effectively modulate microglia as well as infiltrating macrophage population such as MHCII^high^ Kolmer cells [70]. Furthermore, it has been suggested that treatment with oxidized mannan conjugated murine myelin oligodendrocyte glycoprotein 35-55 (OM-MOG35-55) has been shown to induce T-cell tolerance after an increase in expression of M2 markers in MOG-EAE mice [171].

### 7.4. MHCII Modification and Decrease in APC Function

Glatiramer acetate (GA), often mentioned as Cop-1, is a synthetic molecule that resembles MBP_82-100_ and regulates the immune response by direct binding and modification to the MHCII [172]. MBP antagonism has been reported to primarily lead to increased T_h_2 activity in the CNS and shift to anti-inflammatory phenotypes by DCs and macrophage-like cell populations [173,174]. Within its field of direct effects, experiments on THP-1 cells (human leukemia cell line) revealed GA’s ability to inhibit IFN-γ activation [175]. Based on such evidence, it is only natural to discuss how the diverse DRB1 allele signature and the altered structure of the trimolecular complex could distort the landscape about GA effectiveness positively or negatively. An older study described higher binding affinity of GA to P4 pocket of *DRB1*15* as opposed to other structures like DR1 and DR4 [32]. Despite its groundbreaking results in RRMS and PPMS patients, some of GA’s functions remain controversial, with many suggesting that MHCII modulation may not be the only pathway of activity. A study conducted in 2018 showed significant evidence that recognition and interaction with PIR-B (paired Ig-like receptor B) played an important role in the immune response after GA treatment [176]. Finally, it is worth mentioning the example of Dimethyl Fumarate (DMF), the effectiveness of which relies on the activation of transcription of the NFE2L2 gene and its protein product Nrf2 [177]. Its indirect effects have been shown to include reduction of IL-12 and IL-6 expression and a decrease in MHC II, CD80, and CD86 resulting in inhibition of DC maturation and antigen presenting functions [178]. Attenuated IFN-γ signal and downregulation of NF-κB signaling may suggest a similar yet less significant result in some macrophage-like populations (Table 1).

Bruton tyrosine kinase inhibitors (BTKi) are a group of drugs that affect both innate and adaptive immunity and adopt different characteristics regarding selectivity, ligation patterns, and CNS penetrance. Due to their ability to pass effortlessly through the BBB, they have been associated with the suppression of pro-inflammatory activation of resident microglia via Fc-γ receptor, and the regulation of lymphocyte infiltration, leptomeningeal infiltration, and demyelination in MS [179,180,181]. They have also been shown to control phagocytosis and synaptic structure damage in AD and decrease activation levels of microglia and B-cells in NMOSD [182,183]. By promoting repair in highly active inflammatory cells, BTKi may play a strong role in regulating smoldering lesions and the types of MS that regard them, namely PPMS, SPMS, and age related LOMS [184,185].

**Table 1 ijms-25-07354-t001:** The impact of drugs activity in antigen presentation regarding MHCII function and possible HLA allele involvement in effectiveness. The table includes functional details on six commonly applied MS treatments regarding MHC II modulation and antigen presentation, APC polarization, cytokine regulation, migration pattern alteration and immunophenotype dependence.

Drug	Main Function	Impact in Antigen Presentation	References
Glatiramer Acetate	MBP_82-100_ epitope antagonism	- T_h_2 response induction - DC/MΦ polarization * - Affected by HLA structural domain	[172,173,174,175]
Interferon B1	Pro-inflammatory cytokine regulation	- Decrease of active APCs * - Genotype dependent drug response	[39,155,164]
Dimethyl fumarate	Increased Nrf2 transcription	- Decreased DC maturation * - MHCII downregulation	[177,178]
Fingolimod	S1P agonism	- Decreased traffic through the BBB - Impaired myeloid cell activation	[169,186,187]
Natalizumab	VLA-4 (CD49d) integrin targeting	Decreased migration and APC function of MΦs/DCs	[188,189]
Alemtuzumab *	Fc mediated ADCC and CDC of CD52^+^ cells	IL-7 dependent MHC II decrease	[163,190,191]

* Polarization shift and activation profile regards CD1a, and IL-10 signature for DCs, and IL-6, TNF-a and IL-1b for macrophage-like cells.

### 7.5. B-Cells

HLA DRB expression predominates on numerous B-cell surfaces that are proven to contribute to the pathogenesis of MS, namely peripheral memory B-cells, Plasma cells, B-cell follicles in the CNS, B-regs, and generally B-cell populations with antigen presenting properties [192,193]. In MS patients, MHC II rich B-cell populations are primarily found in the meninges and perivascular locations in the CNS, and in smaller numbers in the parenchyma and cytokine profiles of B-cells that are highly associated with both active and inactive lesions [194]. However, little evidence supports significant allelic impact on B-cell function and a possible link with disease outcome. B-cells express a significant amount of MHC II found in the site of neuroinflammation but their antigen presenting functions differ from those found in myeloid cells. APCs, like macrophages and DCs, internalize antigens in a random manner while B-cells only interact with specific targeted proteins through BCR activation and present them via MHCII. However, their function as APCs is undoubtedly selective, as deletion of MHCII in those cells triggers resistance against the disease in mice [195]. This is further supported by the effectiveness of anti-CD20 drugs such as ocrelizumab and ofatumumab, which leads to the reduction of pro-inflammatory Th-cell mediated effects in patients with MS [192,196]. Due to the abundance of surface co-stimulatory molecules that B-cells lack, myeloid APCs seem to contribute to the creation of demyelination hotspots rather than the retention and escalation of neuroinflammation. On the same axis, allelic differentiation of DRB1 molecules may interfere with generalized DRB1 expression levels in a similar manner to those of macrophages, further sustaining the disease inflammatory symptoms. Current pharmacological approaches regarding B-cell population control involve depletion of certain active phenotypes, which might be unnecessary in many cases. Anti-CD20 treatment seem to deplete circulatory CD20^+^ T-cells, though their role in disease progression seems to not be substantial as anti-CD19 therapies have also been proven effective [197,198]. Successful targeting of the trimolecular complex (e.g., via DRB phenotype) may lead to comparable results with less extensive side effects.

Another matter of interest involving B-cells is that *DRB1*15:01* serves as a co-receptor for EBV. EBV infections, recognized as a potent environmental risk for MS, intricately tie with the most prominent genetic risk, *DRB1*15:01* [26,199]. Ectopic lymphoid follicles in the meninges of MS patients can serve as reservoirs for EBV that can facilitate inflammatory migration patterns to traverse throughout the CNS [200]. Yet, from a molecular standpoint, the increased risk attributed to *DRB1*15:01* stems not only from its functional characteristics (e.g., structural affiliation of the trimolecular complex) but rather from its association with generally elevated HLA DR expression in demyelination hotspots. This phenomenon primarily affects cells belonging to the innate immune system, as B-cell development and function are predominantly regulated by separate mechanisms.

## 8. Discussion

The effect and role of CNS neuroinflammation in demyelination and MS is not under debate. Increased pro-MHC II activity and inflammatory markers are undoubtably both an expected symptom of MS and an indication of APC’s homeostasis malfunction. The potential proof of a direct and significant connection between those aspects and HLA phenotype in MS pathophysiology is a matter of great importance in our field of study.

To summarize our knowledge, HLA alleles and high-risk epitopes ultimately owe their notoriety to two major features: (1) Their structural polymorphism and increased enzymatic affinity to MBP peptides (and other autoantigens) and (2) their ability to regulate the innate MHCII expression levels positively or negatively. While the second could be an indirect result of the first, the downstream effects in cell population fluctuations have been repeatedly proven to be a driving factor for MS development, acquiring equal importance in the fight against the disease. Targeting further down the molecular pathway has been a successfully applied practice for a variety of pharmaceutical approaches, as we discussed earlier. However, some suggest that this may not be the case for HLA as the possibility of cross reference and indirect activation is always there. New studies arise that doubt the importance of DRB1 interference in many stages of MS-related neuroinflammation and the progression of the disease but not the onset. A 2020 study reported no difference in myelin responsiveness nor in myelin peptide-reactive precursor cells presence in carriers and non-carriers of *DRB1*15:01* or other HLA risk alleles [201]. On the very opposite side, a recent collaboration of our Research Immunogenetics Laboratory has shown that HLA-DR15 positive humanized mice exhibited significant infiltration of CD8^+^ and CD4^+^ T cells through the CNS barriers and parenchyma. Furthermore, spontaneous and inducible CD8T cell lesions were observed in the brain and spinal cord of HLA-DR15 mice. This robustly enhances and expands our previous knowledge on the core role of this HLA allele on CNS demyelination in humanized animal models [202].

Following the description of a rapidly leaking CNS that characteries MS, an old yet precious study comes to complete the picture about macrophages, HLA genotype, and genetic risk. In their study about MBP_85–99_ adoption, Krogsgaard et al. describe that in the healthy CNS, MBP seems to be steadily produced at low levels by macrophages and microglia in presence of the *HLA-DRB1*15:01* allele [203]. Based on this evidence, a theory is proposed, suggesting that this initial signal is the first necessary step for the development of autoimmunity. During the second step, the imminent diffusion of T-cells into the CNS allows early antigen-presentation to take place, while in the third step, overactivation and rapid re-supply of M1 macrophages into the inflammation site leads to the sustenance and amplification of the initial signal to a significant extent that ultimately leads to advanced autoimmunity.

Numerous population studies and meta-clinical assessments over the past years have proven the assumption that HLA allele diversity has a direct impact in disease outcome in patients with MS, but the defined biochemical signature of those alleles remains unclear. While newer studies try to unravel the molecular connection between known risk alleles and set points of the immune system, evidence seems to indicate a strong relationship between some high-risk alleles and unbalanced macrophage homeostasis. This paradigm is not at all a novel perspective. A shift in the equilibrium between APC states, especially macrophages, has been shown to play a role in the pathophysiology of various diseases. M1 macrophages seem to fuel pro-inflammatory cytokine production, over-stimulation of T cells, and autoantigen destruction such as MBP, while M2 macrophages emerge as pivotal in dampening inflammation and promoting tissue repair, often creating havoc when they increase pathologically, like in the example of fibrotic diseases [204]. Looking deeper within the M2 spectrum, the functional role of the M2c subset might shed some light. M2c and FTY720 (fingolimod), a viable drug for MS, share a key component: S1P. While fingolimod utilizes S1P to attenuate inflammatory cytokine profiles, M2c macrophages express SPHK1, producing active S1P, and independent M2c increase (rather than general M2 boost) has been speculated to be associated with resolution of neuroinflammation [127,205]. To this day however, studies in M2 population dynamics do not yet specify the conditions that characterize MS to confirm or disprove this assumption.

The genetic dimension introduced by HLA-DRB alleles serves to further complicate this already intricate narrative. The revelation that specific alleles, such as *DRB1*15:01*, contribute to a pro-inflammatory microenvironment through the activation of APCs and CD4^+^ T cells suggests a potential therapeutic avenue. Equally intriguing is the possible association of other alleles, such as *DRB1*04:02*, with the induction of M2 macrophages and consequent immune modulation. This allele-specific macrophage-T cell axis offers a fresh perspective on the heterogeneity observed across MS patient populations. So, can HLA-associated modulation of macrophage polarization be an adequate reason for the increased possibility of MS onset? Yes, if a positive connection is proven in rest conditions and naïve cell behavior. Monocytes to macrophages activation is not the same as modulation of already polarized cells. Van Drongelen et al.’s experiments showed that the presence of a potent DRB1 allele can modify the response of an entire cell population when interacting with specific cytokines. Moreover, the modified limits of those cells can play an important role later in the pathophysiology and development of the disease. In RRMS, remission checkpoints resemble the initial onset event. Therefore, a modulated behavior of phenotypic polarization in CNS macrophages and possibly microglia can have the potency to affect remission times depending on the genotype of the patient.

The possible connections between HLA-DRB alleles, TCR components, and downstream immunological pathways warrant meticulous investigation. The orchestration of T-cell responses by HLA alleles, especially through their effects on interactions of the trimolecular complex unlocks a new dimension in understanding the immunopathogenic mechanistic underpinnings of MS. Deeper exploration into these interactions, potentially through integrative approaches spanning genomics and immunology, may uncover the elusive triggers of MS onset and progression. Alterations in MHC II presence in important cell populations (e.g., active lesion perivascular macrophages) could further push the balance towards pathological states that exhibit stronger phagocytic and antigen presenting properties resulting in a so-called “high risk” phenotype.

## 9. Concluding Remarks

This narrative review aims to highlight the crucial role of macrophages and HLA class II alleles in the development of MS. Available literature suggests that risk-labeled alleles seem to have more to offer than a clinically suspicious landscape. The revelation of molecular mechanisms that depend on the alteration of structural domains on the peptide products as well as the general MHC II expression by some cell types uncover critical new knowledge around the disease’s pathophysiology and strategies of treatment. Unfortunately, the molecular steps and cross-pathing involved in a single immune response are countless. On the one hand, the allelic ability to modulate affinity in the paradigms of *DRB1*15*, *DRB1*04*, and *DRB1*01* set a foundation for onset of autoimmunity. However, a fair number of risk alleles that are notoriously reported to influence the clinical condition of MS patients, such as DR13, remain in the dark in terms of their molecular basis. On the other hand, the vessels that promote these alleles and make them abundantly available in the pathological CNS, namely MHC II rich APC populations, seem to play a comparably high role both in onset and progression of MS. Upstream dysregulation of the population homeostasis on certain macrophage subtypes result in concentrated or even systemic deterioration of the demyelination profiling by encompassing pathways previously unknown yet sufficiently targeted indirectly by commercially available therapeutics. Population dynamics in neuroinflammatory APCs, though the prism of immunogenetics, is a topic largely unexplored and we believe that it holds great potential for future work in the topic of demyelinating autoimmune diseases and the field of immunopharmacology. Further emphasizing such immune mechanisms is imminent as the need for more personalized approaches arises.

## Figures and Tables

**Figure 1 ijms-25-07354-f001:**
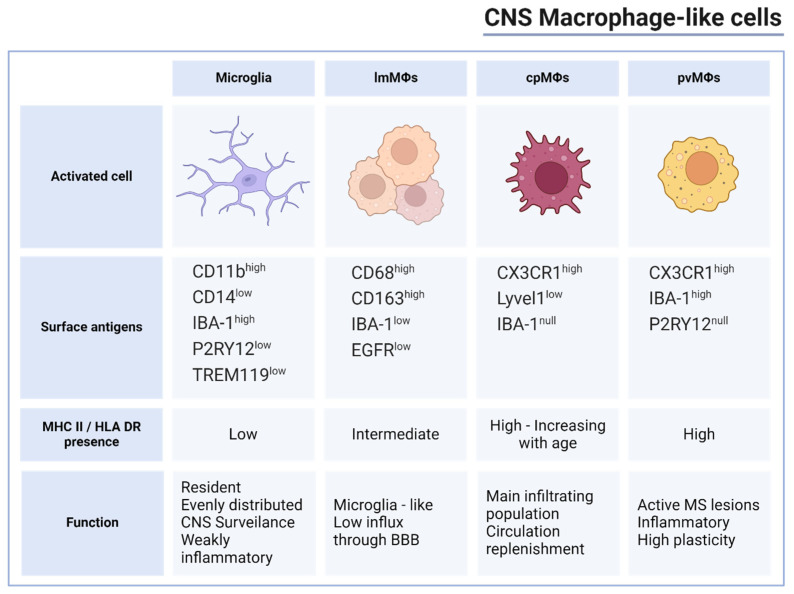
Macrophage subsets in MS and presence of MHC II. Summarization of the distinct features of the 4 basic macrophage-like populations in the CNS including surface antigens used experimentally for FACS analysis and cell-shorting techniques, MHC II and HLA-DR surface presence along with relevant antigen presenting properties as well as basic functional characteristics of each population regarding involvement in inflammation, replenishment, plasticity, and tissue distribution. (Created with www.biorender.com, accessed on 14 June 2024).

**Figure 2 ijms-25-07354-f002:**
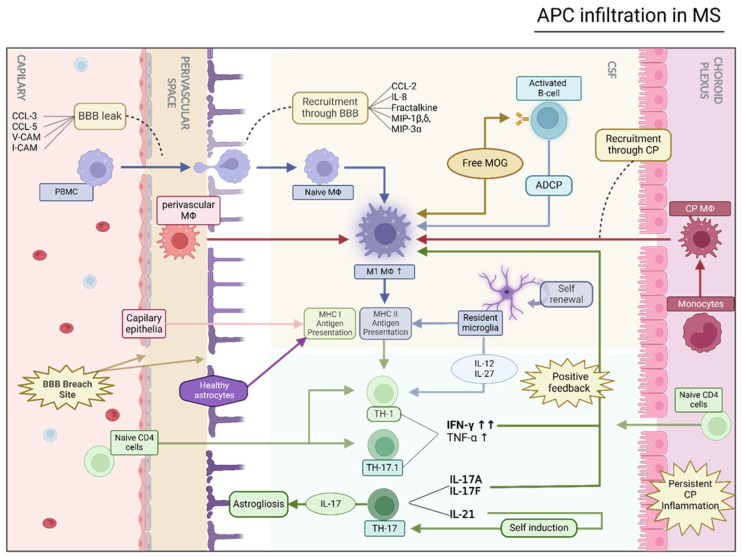
APC infiltration and recruitment through the BBB in MS. This figure displays the functional pathways of the inflammatory CNS as follows. Blood-brain barrier (BBB) structure and leak. The BBB is displayed with its main components namely endothelial cells, tight junctions, and perivascular space. Disrupted endothelial borders and astrocyte integrity allow immune cells to infiltrate the CNS parenchyma. The choroid plexus is depicted with its epithelial barrier and associated macrophages as well as the leakage points. Infiltration patterns are focused on the axis of macrophage overactivation and possible feedback reactions within them. CP-CNS and BBB-CNS leak is illustratively displayed with different colors, symbols, and annotations. (Created with www.biorender.com/, accessed on 14 June 2024).

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
