# Peer review of "Macrophages and HLA-Class II Alleles in Multiple Sclerosis: Insights in Therapeutic Dynamics"

_ijms, 2024, doi:10.3390/ijms25137354_

Round 1

Reviewer 1 Report

Comments and Suggestions for Authors

In the current review paper, the authors further explore and summarize the role of various cells of the immune system (T-cells, macrophages, and microglia) during the inflammation present in the course of MS. I consider it particularly important to discuss in detail the current state of knowledge on the molecular mechanisms underlying BB barrier penetration and also to attempt to systematize knowledge on the pharmacological use of the findings. I consider the topic of the paper to be current and clinically important. The meticulous analysis of the issue and the excellent quality of the informative figures deserve special recognition.

Specific comments:

Line 108 - please replace with "TNF-α"

Line 120- the abbreviation ADEM is not used once in the paper. What is the point of introducing it?

Line 188, 652 - correct to "neuromyelitis optica". I see no justification for writing the name of the disease in capital letters.

Line 257, 391 - please provide information on BioRender™ software (license number, access, etc.).

Line 652 - the abbreviation NMSOD has already been introduced on page 69.

Author Response

Thank you for revising our article and providing valuable feedback. In the attached revised file, we have included the following changes, highlighted in yellow color which is separate for each reviewer’s changes.

Comment 1: Regarding Biorender info, we have followed citation guides as they are presented on the software site. License for both figures were included in the zip folder during the submission. If you still believe that such info can be proven useful in the text, then we are more than happy to include it according to your indications.

Comment 2: Regarding current knowledge on BBB penetration mechanisms, we incorporated a small paragraph in the respective section mentioning some basic characteristics. However, the goal of our study is not to thoroughly describe the mechanisms involving BBB function but only those regarding the immune complex that characterizes MS. We hope that it serves the purpose of your suggestion.

Comment 3: Regarding the pharmacological findings, you mentioned that we attempt to systematize the knowledge but unfortunately, we are struggling to find a way to do this. In order to provide a visual summary of the pharmacological approaches that we mentioned, we created the table, so that the reader can easily understand our core points. We could write a paragraph at the end of the section summing up the functions of each drug, shorting them according to them, if you believe it would be useful.

Please excuse us if in any case we have misunderstood any of your suggestions, and we look forward to hearing from you soon.

Reviewer 2 Report

Comments and Suggestions for Authors

Prapas and Anagnostouli performed a narrative review of the macrophages and HLA-class II alleles in multiple sclerosis.

It is an interesting manuscript and provides a deep description of the APC and suggestive pathways related to MS. I would only advise including at least a paragraph smoldering MS and how it differs from our current understanding of the pathophysiology of MS.

Author Response

Thank you for revising our article and providing valuable feedback. In the attached revised file, we have included the following changes, highlighted in green color which is separate for each reviewer’s changes.

We incorporated a small paragraph discussing smoldering lesions and the characteristics of the macrophages involved. We also mentioned the effect of BTK inhibitors’ effect in those lesions in the Pharmacological Approaches section.

We are looking forward to hearing from you and receiving your comments.

Reviewer 3 Report

Comments and Suggestions for Authors

The authors reviewed the literature on genetic alterations of macrophages in multiple sclerosis. The authors should follow my comments in the attached pdf file to make the article easier to understand for readers.

Comments on the Quality of English Language

Author Response

Thank you for revising our article and providing valuable feedback. In the attached revised file, we have included the following changes, highlighted in turquoise blue color which is separate for each reviewer’s changes. We conducted a proof-reading throughout the manuscript, and we hope that the changes we made are satisfactory. We are looking forward to your response.

[comment 1] Please add a comprehensive list of abbreviations to the manuscript. Also, define each abbreviation the first time it is used in the text.

[response 1] We added a glossary at the end of the manuscript including all abbreviations used

[comment 2] Please shorten all run-on sentences throughout the manuscript

[response 2] We attempted to shorten run-on sentences in the manuscript and corrected grammatical errors in the following lines:

73: of BV-2

108: result in

109 of separate

111: within the

247: spotlight

[comment 3] Break this paragraph into separate paragraphs.

[response 3] Paragraph has been broken into two separate paragraphs

Reviewer 4 Report

Comments and Suggestions for Authors

The article provides a comprehensive review of the role of macrophages and HLA class II alleles in the pathogenesis of multiple sclerosis (MS). The authors cover several important aspects, including HLA risk alleles, macrophage ontogeny, antigen presentation, and pharmacological approaches targeting these pathways.

However, there are a few areas that could be improved:

  1. The manuscript would benefit from a more concise introduction that clearly states the main objectives and the significance of the review.
  2. Some sections, such as the discussion on epigenetic modulation, could be more extensively developed to provide a more comprehensive understanding of the topic.
  3. The authors might consider including a figure that illustrates the key pathways and interactions discussed in the review to help readers better visualize the complex relationships.
  4. The conclusion section could be strengthened by summarizing the main points and highlighting the potential implications for future research and clinical applications.
  5. A thorough proofreading is recommended to correct minor typographical and grammatical errors throughout the manuscript.

Overall, this is a well-written and informative review that contributes to our understanding of the complex interplay between macrophages, HLA alleles, and MS pathogenesis. With some minor revisions, this manuscript has the potential to be a valuable resource for researchers and clinicians interested in MS and neuroimmunology.

Author Response

Thank you for revising our article and providing valuable feedback. In the attached revised file, we have included some of the changes that you suggested, highlighted in pink color which is separate for each reviewer. Commenting on your suggestions we reply the following:

[comment 1] The manuscript would benefit from a more concise introduction that clearly states the main objectives and the significance of the review.

[comment 4] The conclusion section could be strengthened by summarizing the main points and highlighting the potential implications for future research and clinical applications.

[response 1 and 4] We tried to enrich the introduction and conclusion as you indicated while trying not to increase the total word count too much. We hope that those changes are satisfactory and achieve the goal of improving the readers’ ability to comprehend the points of the article.

[comment 2] Some sections, such as the discussion on epigenetic modulation, could be more extensively developed to provide a more comprehensive understanding of the topic.

[response 2] We intentionally tried to not elaborate too much on the epigenetics section, as we are planning to publish another review regarding HLA and epigenetics in MS in the course of the next year. We also wanted to keep the word count low so that it increases readability, and we have the freedom to be more thorough with some other sections. However, we added some extra information on the section that we believe is worth mentioning and provide the reader with a more detailed description of the topic (methylation in and out of the MHC locus). If you insist on expanding the section, we are more than happy to do so.

[comment 3] The authors might consider including a figure that illustrates the key pathways and interactions discussed in the review to help readers better visualize the complex relationships

[response 3] Indeed, a figure regarding key pathways and interactions such as spontaneous ligation of free MBP, illustration of structural differences etc. would be very positive for the reader to understand those mechanisms. We will have to kindly ask for some more time in order to incorporate such an addition as it requires some focused work and planning.

[comment 5] A thorough proofreading is recommended to correct minor typographical and grammatical errors throughout the manuscript.

[response 5] Following We conducted another proofreading throughout the manuscript and corrected many grammatical errors. We hope that the revised text meets the standards of the journal.

Again, we thank you for your comments, we appreciate it, and we are looking forward to hearing from you.
